# High genome plasticity and frequent genetic exchange in *Leishmania tropica* isolates from Afghanistan, Iran and Syria

**Hedvig Glans**[1,2]*, **Maria Lind Karlberg**[3], **Reza Advani**[3], **Maria Bradley**[2,4], **Erik Alm**[5], **Björn Andersson**[6], **Tim Downing**[7]

**1** Department of Infectious Diseases, Karolinska University Hospital, Stockholm, Sweden, **2** Division of Dermatology & Venereology, Dept of Medicine Solna, Karolinska Institutet, Stockholm, Sweden, **3** Department of Microbiology, The Public Health Agency of Sweden, Stockholm, Sweden, **4** Department of Dermatology and Venerology, Karolinska University Hospital, Stockholm, Sweden, **5** The European Center for Disease Prevention and Control, Stockholm, Sweden, **6** Department of Cell & Molecular Biology, Karolinska Institutet, Stockholm, Sweden, **7** School of Biotechnology, Dublin City University, Dublin, Ireland

* hedvig.glans@ki.se

**Data Availability Statement:** The sequences are available through ENA (SRA accession PRJEB45563) and Zenodo (https://doi.org/10. 5281/zenodo.5645033, https://doi.org/10.5281/

## Abstract

### Background

The kinetoplastid protozoan *Leishmania tropica* mainly causes cutaneous leishmaniasis in humans in the Middle East, and relapse or treatment failure after treatment are common in this area. *L. tropica*'s digenic life cycle includes distinct stages in the vector sandfly and the mammalian host. Sexual reproduction and genetic exchange appear to occur more frequently than in other *Leishmania species*. Understanding these processes is complicated by chromosome instability during cell division that yields aneuploidy, recombination and heterozygosity. This combination of rare recombination and aneuploid permits may reveal signs of hypothetical parasexual mating, where diploid cells fuse to form a transient tetraploid that undergoes chromosomal recombination and gradual chromosomal loss.

### Methodology/principal findings

The genome-wide SNP diversity from 22 *L. tropica* isolates showed chromosome-specific runs of patchy heterozygosity and extensive chromosome copy number variation. All these isolates were collected during 2007–2017 in Sweden from patients infected in the Middle East and included isolates from a patient possessing two genetically distinct leishmaniasis infections three years apart with no evidence of re-infection. We found differing ancestries on the same chromosome (chr36) across multiple samples: matching the reference genome with few derived alleles, followed by blocks of heterozygous SNPs, and then by clusters of homozygous SNPs with specific recombination breakpoints at an inferred origin of replication. Other chromosomes had similar marked changes in heterozygosity at strand-switch regions separating polycistronic transcriptional units.

zenodo.5647430). The supplementary data, valid VCF files, phylogeny files, read coverage per chromosome, de novo assemblies, PacBio assembly, read coverage per chromosome for de novo assemblies, homozygous and heterozygous SNPs per chromosome, locus-specific somy levels per sample, read-depth allele frequencies per chromosome per sample, and R code scripts used for analyses are publicly available on FigShare at https://figshare.com/projects/Leish_tropica_2021/118014.

**Funding:** The author (BA) has recived funding from Vetenskapliga rådet/Swedish Research Counsil, 2016-02951 (https://www.vr.se/english.html). The funders had no role in study design, data collection and analysis, decision to publish, or preparation of the manuscript.

**Competing interests:** The authors have declared that no competing interests exist.

## Conclusion/significance

These large-scale intra- and inter-chromosomal changes in diversity driven by recombination and aneuploidy suggest multiple mechanisms of cell reproduction and diversification in *L. tropica*, including mitotic, meiotic and parasexual processes. It underpins the need for more genomic surveillance of *Leishmania*, to detect emerging hybrids that could spread more widely and to better understand the association between genetic variation and treatment outcome. Furthering our understanding of *Leishmania* genome evolution and ancestry will aid better diagnostics and treatment for cutaneous leishmaniasis caused by *L.tropica* in the Middle East.

## Author summary

Cutaneous leishmaniasis is mainly caused by *Leishmania tropica* in the Middle East, where it is known for treatment failure and a need for prolonged and/or multiple treatments. Several factors affect the clinical presentation and treatment outcome, such as host genetic variability and specific immune response, as well as environmental factors and the vector species. Little is known about the parasite genome and its influence on treatment response. By analysing the genome of 22 isolates of *L. tropica*, we have revealed extensive genomic variation and a complex population structure with evidence of genetic exchange within and among the isolates, indicating a possible presence of sexual or parasexual mechanisms. Understanding the *Leishmania* genome better may improve future treatment and better understanding of treatment failure and relapse.

## Introduction

Leishmaniasis is a vector-borne parasitic disease transmitted by sand flies. At least twenty *Leishmania* species are pathogenic to humans and can cause a spectrum of clinical manifestations, from chronic local ulcers to cutaneous leishmaniasis (CL), and to infection of internal organs in visceral leishmaniasis [1–3]. Host genetic variability, host immune responses, sand fly feeding behaviour, environmental factors and parasite species and strain variation all influence the clinical manifestation and the outcome of the infection [4]. Genome sequencing within *Leishmania* species has mostly shown a lack of a clear association between genetic differences and the clinical pathology [5].

   *L. tropica* is a heterogeneous species complex with a broad geographical distribution across Africa and Eurasia [3,6,7] that causes mainly CL, though visceral leishmaniases has been reported [8,9]. *L. tropica* is typically an anthroponotic disease, though zoonotic transmission may be possible as well [10–12]. Although the burden of leishmaniasis is decreasing globally, regions with *L. tropica* have increasing rates due to local conflicts and associated population displacement, migration and relocation [3], which exacerbates incorrect CL diagnoses and inadequate access to appropriate healthcare and treatments. CL caused by *L. tropica* in the Middle East can be difficult to treat: lesions may relapse (called leishmaniasis recidivans) and multiple or prolonged treatments are often necessary [13,14].

   *Leishmania* have specific *Phlebotomine* sandfly vector compatibilities that affect transmission [15,16]. In most geographic areas, *L. tropica* is transmitted by its most common vector, the female *Phlebotomus sergenti* sandfly [17]. Although other vectors like *P. arabicus* in

northern Israel [18] and *P. guggisbergi* in Kenya [19] also are known. *Phlebotomus spp.* can be infected by different *Leishmania spp.*, including *P. perniciosus* [20], *P. tobbi* by *L. tropica* and *L. infantum* [20,21] and *P. guggisbergi* by *L. tropica* and *L. major* [19]. Hybrids capable of transmitting have formed in all these vectors [22]. *L. tropica*'s digenic life cycle has an extracellular promastigote stage where they occupy different regions of the alimentary tract of the vector, and subsequently as amastigotes within mammalian macrophages [23]. During the extracellular growth and development in the vector, *Leishmania* is capable of non-obligatory meiotic genetic exchange [24].

Evidence of hybridisation has been observed in diverse natural isolates of *Leishmania* [25–28] and in culture [29]. It is possible but still uncertain that reproduction in *Leishmania* could be facilitated by genes with high similarity to ones involved in meiosis [24] to allow crossing-over, resulting in recombinant chromosomes [30]. As more *Leishmania* are genetically profiled, the precise mechanisms involved and frequencies of *Leishmania* meiotic or quasi-sexual events are beginning to become clearer. Classical meiosis like that of the related kinetoplastid *Trypanosoma brucei* [31] has not been observed in *Leishmania*-infected sand flies, but there is indirect evidence that it does occur [32]. Although the universal mosaic aneuploidy of *Leishmania* is inconsistent with classical meiosis, experimental backcrosses can produce hybrids in sand flies with varying degrees of genetic relatedness among parental lines [32]. Possible alternative mechanisms like parasexual reproduction have been considered for *Leishmania* based on observations in *Saccharomyces* [32], such as a tetraploid meiotic cycle in which diploid parental cells fuse followed by a meiosis, resulting in diploid F1 [32]. Another model is parasexual reproduction in which a fusion of parental cells is followed by karyogamy, re-shuffling of DNA regions, and gradual loss of chromosome copies during mitoses [24,29,30,33]. There is evidence for this in experimental *L. major* and *L. infantum* crosses [34] and *Trypanosoma cruzi* [31,35]. These mechanisms are not mutually exclusive, and different types of genetic exchange are possible [35,36].

*Leishmania* hybrids generated *in vitro* and in sand flies are mostly diploid at the time of culturing and contain approximately equal amounts of genetic material from both parental strains [32,37], suggesting meiosis-like processes. Triploid and tetraploid hybrid offspring have also been observed, which could indicate other mechanisms [24,29,37]. Genome sequencing of triploid hybrids indicated that a triploid F1 (3n) may be a product of fusion between a diploid cell that failed to undergo meiosis (2n) and a haploid cell (1n) [37]. Hybrid strains have been generated experimentally with different *Leishmania spp.* in sand flies, both within and between species [27,32,38–40]. Natural hybrid isolates have been found in a range of geographical settings, suggesting abundant natural genetic exchange within and between different species that is important for the evolution of the parasite [25–27,41–43] to enhance transmission potential and increase the fitness of the parasites [29,32,34,44]. *Leishmania* genomes are unstable and lack promoter-dependent gene regulation [45]. Together with post-transcriptional regulations, *Leishmania* exploits gene dosage by chromosome and gene copy number variation [46–48] which are driven by variable environments that maintain expression levels and genetic diversity [49]. Thus, inter-species hybrids may have facilitated the spread of *Leishmania* to new geographic regions through changes in vector specificity [16]. Although new interspecies hybrids are usually not able to carry out genetic exchange, intra-species hybrids could produce new progeny. The ability to produce hybrids *in vitro* and *in vivo* varies between species: it is lower in *L. major* [24,37] than *L. tropica*, where a higher frequency of hybrid formation in co-infected sand flies has been observed [32].

*Leishmania* possesses universal aneuploidy in natural isolates, laboratory strains, in experimental hybrids and during culture [32,50–52]. This mosaic aneuploidy is primarily a result of mitotic asymmetric chromosome allotments [30]. This enhances the genomic variation

through differences in gene copy numbers and chromosomal duplication events [47,53]. Aneuploidy may facilitate eliminating deleterious mutations during chromosome loss, and the persistence of an intra-strain genetic heterogeneity may be beneficial for the cell population [54].

In *Leishmania*, there may be only one single region per chromosome where DNA replication initiation occurs, detected during the S phase [55]. This may lead to incomplete duplication of the larger chromosomes during S phase and an inadequate duplication of the genome prior to cell division [55]. *Leishmania* employs sub-telomeric DNA replication beyond S phase, which may be less effective in maintaining genome integrity, and in this way contributes to variability and aneuploidy [56].

Gene expression in *Leishmania* is carried out through polycistronic transcriptional units (PTUs) [57] separated by strand-switch regions (SSRs) where RNA polymerase II can transcribe bidirectionally [58]. Genes are transcribed as multigene pre-mRNAs with a single constitutive transcription start site (TSS) [58,59] and are trans-spliced to form mature mRNAs. This means gene dosage is crucial [46, 60] since gene regulation is mostly post-transcriptional [61,62], leading to both intra- and extrachromosomal genome-wide gene copy number variation and mosaic aneuploidy. *L. tropica* has high levels of allelic diversity and heterozygosity, consistent with frequent full genome-hybridization, most likely due to natural outcrossing [32,63].

Despite previous research in the field, much remains unclear regarding the diversity, evolution, and genetic exchange of *L. tropica*. In this study, we have investigated 22 isolates of *L. tropica* from 21 patients, focusing on the genetic diversity within and between the isolates and correlating the results to their geographic sources in Afghanistan, Iran and Syria. The study revealed large-scale intra- and inter-chromosomal changes in diversity driven by recombination and aneuploidy that enhance our understanding of *Leishmania* genome evolution and ancestry.

## Methods

### Ethics statement

Ethical approval was obtained from the Central Ethical Review Board in Stockholm (2015/2162–31). Informed consent was not found to be necessary, as the samples, included in the genome study, and the clinical data were anonymized, and only isolated parasites were studied.

### Sample collection

The *L. tropica* isolates were sampled from 21 patients diagnosed with CL in 2007–2017. The patients were from Syria (17), Afghanistan (3) and Iran (2). The patients were treated with sodium stibogluconate (7), cryotherapy (5), liposomal amphotericin (5), meglumine antimoniate (2), fluconazole (1), cryotherapy and sodium stibogluconate (1), cryotherapy and liposomal amphotericin (1), or had no treatment (2) (Table 1). One patient with leishmaniasis recidivans had not visited endemic areas since first symptoms of CL, was first sampled in 2014 followed with a second sample in 2017 after ineffective sodium stibogluconate treatment, resulting in 22 *Leishmania* genomes for investigation.

### Culturing, DNA extraction, library preparation and short read sequencing

The isolates were stored at -156˚C at the Public Health Agency of Sweden. Promastigotes were cultivated in RMPI 1640 medium with L-Glutamin, HEPES, Penicillin-Streptomycin and fetal

**Table 1. 21 patients with 22 isolates of *L. tropica* (\* from the same patients) infected in the Middle East (three from Afghanistan, two from Iran, 17 from Syria).** Ten had received treatment prior sampling (nine SSG, one cryotherapy) and twelve were cured on first-line treatment after sampling. Six were cured on second-line treatment (3 LA, 3 SSG il). One healed by itself and two were lost after sampling. One patient continued to relapse regarding treatment. \*\* healed by itself. SSG stands for sodium stibogluconate, LA for liposomal amphotericin, MA for meglumine antimoniate, il for intralesional, im for intramuscular, iv for intravenous, cured for absence of clinical relapse for 6 months after treatment, relapse for recurrence of a lesion after the lesion had healed without any known new exposure to the parasite, treatment failure for absence of clinical signs of re-epithelialisation in the lesion during or within two months after treatment.

| Sample ID | Country | Previous treatment | Treatment | First-line treatment | Treatment outcome | Second-line treatment | Treatment outcome |
|---|---|---|---|---|---|---|---|
| 07_00242 | Iran | Yes | SSG | No treatment | Cured | | |
| 07_01513 | Syria | No | | NA | | | |
| 13_00550 | Syria | Yes | SSG | SSG iv | Cured | | |
| 13_01024 | Syria | No | | Cryotherapy | Cured | | |
| 13_01233 | Afghanistan | No | | Cryotherapy | Cured | | |
| 13_01390 | Syria | Yes | SSG | SSG iv | Relapse | LA | Cured |
| 14_00642 | Syria | No | | SSG iv | Relapse | LA | Cured |
| 14_00771 | Syria | Yes | SSG | LA | Cured | | |
| 14_00849 | Syria | No | | MA im | Cured | | |
| 14_01223* | Syria | Yes | SSG | SSG iv | Treatment failure | LA | Cured |
| 15_00019 | Syria | Yes | SSG | LA | NA | | |
| 15_01088 | Syria | Yes | SSG | SSG il | Cured | | |
| 15_01620 | Syria | No | | LA | Treatment failure | SSG iv | Cured |
| 15_02015 | Syria | No | | MA im | Cured | | |
| 15_02480 | Afghanistan | No | | Fluconazole | Treatment failure | LA | Treatment failure** |
| 15_02576 | Syria | No | | Cryotherapy | Cured | | |
| 15_02597 | Syria | Yes | Cryotherapy | Cryotherapy | Treatment failure | SSG il | Cured |
| 16_00075 | Afghanistan | No | | LA + Cryotherapy | Treatment failure | SSG il | Cured |
| 16_00674 | Syria | No | | LA | Cured | | |
| 16_00964 | Iran | No | | SSG il + Cryotherapy | Cured | | |
| 16_14706 | Syria | Yes | SSG | Cryotherapy | Cured | | |
| 17_01604* | Syria | Yes | | | | | |

calf serum at 23˚C [64]. All isolates were grown for the minimum time necessary to produce adequate parasite numbers to isolate enough genomic DNA (gDNA) for library preparation. A QIAamp Mini Kit (QIAGEN) was used to extract DNA, and Qubit dsDNA BR Assay Kit (Invitrogen, Thermo Fisher Scientific, USA) was used to measure DNA concentrations—all according to manufacturer's protocols. 200 ng gDNA was used for library construction using the Ion Xpress Plus Library Kit following the manufacturer's instruction. The libraries were quantified by an in-house qPCR, MGB (minor groove binder) assay [65]. For template preparation, gDNA libraries were pooled to a final concentration of 30 or 50 pM and clonally amplified using the Ion Torrent Chef system using the Ion 510&520&530 Kit-chef or Ion 540 Kit-chef and then loaded onto an Ion 530 or 540 chip, respectively. Massively parallel sequencing was then performed on an IonTorrent S5 XL instrument (Thermo Fisher Scientific, USA) according to the manufacturer's protocol. The template preparation and sequencing were repeated for each sample until the initial obtained sequencing data corresponded to more than >30x genome-wide coverage.

## Read processing, mapping and SNP screening

Sequencing of the libraries resulted in an average of 8.73±4.40 (mean±SD) million single-end raw reads of up to 557 bp in length, with at least 4.14 million raw reads per sample. The quality of the initial DNA read libraries varied considerably, and careful quality control was conducted

to ensure only high-quality reads were retained and low-quality reads were discarded (S1 Table). The reads were trimmed including removal of adaptor sequence using the Torrent Suite software with standard settings (Torrent suite v5.2.2) (SRA accession PRJEB45563) resulting in 22 valid short read libraries. Remaining low-quality bases with a base quality <30 were removed with the FASTX-Toolkit v0.0.14 (http://hannonlab.cshl.edu/fastx_toolkit/), and the absence of remaining low-quality reads was verified with FastQC v0.11.5 (www.bioinformatics.babraham.ac.uk/projects/fastqc/). These high-quality reads had lengths of 70–557 bases after screening and an average of 6.98±2.58 million reads per sample (S1 Table).

'The reference genome used for analyses was the *L. tropica* LRC-L590 (MHOM/IL/1990/P283) assembly v2.0.2 with annotation from Companion [66] accessed through the Sanger Institute. It was masked using Tantan v13 [67] to exclude repetitive sequences, short tandem repeats, homopolymers and low-quality regions. The screened reads were mapped to this reference with Smalt v7.6 (http://www.sanger.ac.uk/science/tools/smalt-0) with a k-mer length of 19 so that candidate variants could be determined. Initial candidate SNPs (278,616±25,792) were extracted using BCFtools v1.10.2 [68] from which valid SNPs were selected that had base quality >25, mapping quality >30, >4 forward derived alleles, >4 reverse derived alleles, and read coverage >10-fold. This was visualised for each parameter with RStudio v4.0 across the collection (S1 Fig). Regions with 100 bases of the chromosome edges were excluded. This resulted in a total of 301,659 valid SNPs across all samples, with 170,183±63,717 SNPs per library across the genome, with at least 69,843 SNPs per sample (S1 Table). Excluding contigs not assigned to chromosomes, this meant 169,753±63,717 valid SNPs per sample, with an average of 430±186 valid SNPs on contigs. The association between the numbers of initial reads and candidate SNPs was high ($r^2$ = 0.243) compared to the association between the numbers of valid reads and valid SNPs ($r^2$ = 0.028), suggesting that quality-control improved the accuracy of the true variation present. After careful checking, there were 17 mutations at 11 SNPs that were triallelic SNPs where both derived alleles were observed, based on derived allele frequencies <0.9 and read depth >14. SNP density measurements assumed a quasi-Normal distribution across the chromosomes based on Shapiro-Wilk normality tests excluding outlying chromosomes with extreme SNP rates.

### *De novo* assembly

Each read library (except 13_01024) was assembled *de novo* for all k-mer lengths from 33 to 253 inclusive with a step size of 10 bases with Megahit v1.2.9 [69], retaining contigs of >1 Kb. This optimised the assembly using de Bruijn graph information from different k-mers and can cope well with heterogeneous data whose local copy numbers and read depths varied [69]. Here, it was used to verify large structural rearrangements and has been previously used on *Leishmania* genomes [70]. These initial *de novo* assemblies had an average of 8,152±3,025 contigs with an average N50 of 7,816±3,218. They were contiguated using the *L. tropica* reference with ABACAS [71] to produce assemblies whose chromosomes spanned 32.85±0.94 Mb excluding N bases, which was similar to the reference's length (32.93 Mb). A comparison of the Z-normalised reference chromosome lengths to the Z-normalised *de novo* ones for all samples indicated that few chromosomes had extreme differences in lengths after Benjamini-Hochberg (BH) p value correction. However, lengths were longer than expected for chromosome 7 for 14_00771 (p = 0.022) and 15_0215 (p = 0.038), and several other chromosomes noted when discussed below.

### Long read sequencing, assembly and SNP screening

PacBio long read sequencing of 13_00550 produced 1.05 billion bases across 68,806 reads with a mean length of 15,225 bases and a N50 of 33,952. After quality control and mapping, 64,437

high-quality reads resulted in an average of 40-fold coverage across the *L. tropica* reference genome. After assembly and polishing by Hierarchical Genome Assembly Process 3 (HGAP3, [72]), the 13_00550 assembly had 2,655 contigs with a contig N50 of 19,171 spanning 32.7 Mb. This assembly masked with Tantan v13 as above and contiguated using the *L. tropica* reference with ABACAS [71] to produce an assembly whose chromosomes spanned 34,152,472 bp: 29,861,533 bp excluding N bases. This was indexed using Smalt v7.6 and reads for the 13_00550 short read library were mapped to it. After quality control and SNP calling as above using BCFtools v1.10.2, this found 21,663 heterozygous and zero homozygous SNPs across the chromosomes, supporting our SNP calling approach.

## Inference of population structure and ancestry

Population structure was evaluated across the 301,659 SNPs by constructing a nuclear DNA maximum-likelihood phylogeny based on 168,327 alignment patterns using RAxML v8.2.11 with a GTR+gamma substitution model [73]. This was visualised with the *L. tropica* reference as an unrooted phylogeny using R packages ape v5.5 and phytools v0.7–70. To further resolve ancestries within discrete genomic chromosomes and regions, phylogenies per chromosome were constructed as above using RAxML v8.2.11 were visualised with phytools v0.7–70 and dendextend v1.14.0 [74].

We classified the 22 samples into genetically distinct groups using a hierarchical Bayesian clustering (BHC) with a Dirichlet Process Mixture model implemented in FastBAPS v1.0 [75]. This interpreted the SNP variation as a sparse matrix with an optimised hyperparameter of 0.092 for the genome-wide pattern using R packages ape v5.5, ggplot2 v2.3.3.3, ggtree v2.4.1 [76], phytools v0.7–70 and treeio v1.14.3. Isolates were allocated to genetically distinct groups per chromosome using FastBAPS v1.0 with a population mean prior and chromosome-specific optimised hyperparameters. This clustering was repeated for regions of interest on chromosomes 10, 12, 23, 29, 31 and 36.

## Kinetoplast DNA variation

The kinetoplast DNA (kDNA) is a high copy number network of circular minicircle and maxicircle DNA molecules. The 22 libraries were mapped to the masked *L. tarentolae* reference maxicircle kDNA (20,992 bp, accession M10126.1, [77] using Smalt v7.6. The *L. tropica* reference maxicircle was tested, but the assembly quality of that contig limited any inferences. Valid SNPs were determined where the mapping quality > 30, the read-depth allele frequency (RDAF) > 0.8, and the read depth of high-quality reads > 10, excluding sites within 100 bp of the maxicircle reference ends. This found 1,594 SNPs distinguishing all of the 22 isolates from the *L. tarentolae* reference, which phylogenetically clustered most closely with isolate 13_01390. Within the 22 *L. tropica* isolates, there were 220 variable sites, with a frequency per sample ranging from 9 to 154 SNPs with an average of 82±39 SNPs. The variation was visualised with IGV v2.8.12 with the *L. tarentolae* annotation. A phylogeny was constructed as above using RAxML v8.2.11 with a GTR+CAT substitution model [73] and using R packages ape v5.5 and phytools v0.7–70. As above, genetically distinct groups were assigned using FastBAPS v1.0.

## Aneuploidy and structural variation

Chromosomal somy levels were determined using the read depth coverage per base normalised by the median genomic coverage for each library [78] scaled as the haploid depth. This excluded sites at the first 7 Kb or last 2 Kb of chromosomes [79]. All libraries had at least 21-fold median genome-wide coverage except samples 13_01233 and 13_00550 whose median

coverage was 13-fold, so an alternative somy estimation using binning was not essential here. Read depth was determined from the mapping patterns for chromosomal regions using 5 Kb windows to account for local variability. This yielded power to examine large structural changes of >10 Kb: small structural variants and indels were not examined in detail here due to potential artefacts associated with the sequence quality.

## Results

We isolated, sequenced and analysed 22 *L. tropica* isolates from infected patients originally from Afghanistan (n = 3), Iran (n = 2) and Syria (n = 17) to decipher processes driving genome evolution in this unique collection.

### Genomic diversity and population structure in *L. tropica*

The analysis of the sequence data identified 301,659 SNPs across all sample including multi-allelic sites and 1,396 unique SNPs in contigs including multi-allelic sites. The isolates had differences in chromosomal diversity, with a range of 69,662 to 277,195 SNPs per isolate (SNPs on contigs were not examined in this study). Most SNPs were heterozygous: the number of homozygous SNPs per sample was 4,886±2,855, corresponding to 0.15 SNPs/Kb, with a range of 2,346 to 13,371. The number of heterozygous SNPs was much higher at 164,886±64,763 per sample, corresponding to 5.0 SNPs/Kb, with a range of 64,391 to 273,463. The numbers of homozygous and heterozygous chromosomal SNPs per sample had a small negative correlation ($r^2$ = 0.07, S2 Fig), suggesting processes other than genetic drift were contributing to the variation. We validated our careful sequencing, processing and SNP ascertainment approach by long PacBio read sequencing of isolate 13_00550. This resulted in zero genome-wide homozygous SNPs after self-mapping its own short reads to its *de novo* assembly, and high heterozygosity compared to when the reads were mapped to the *L. tropica* LRC-L590 reference genome.

Two genetically distinct groups were evident based on the patterns of a chromosome-wide phylogeny constructed with RAxML v8.2.11 [73] and population structure from FastBAPS v1.0 [75] (Fig 1). The first genetically distinct group (n = 7) was related to the *L. tropica* reference genome and hence is referred to as the reference group. The second (n = 15) was divergent from the reference genome and is here called the non-reference group. Isolates from Syria, Afghanistan and Iran were found in both groups. The two isolates from the same patient in Syria (14_01223 and 17_01604) were both from the non-reference group. Certain isolates with long external phylogenetic branches that were phylogenetically close to being between the two groups had instances of inconsistent group assignment by FastBAPS. For example, 12 out of the 36 chromosomes of 16_00964 were allocated to the reference group by FastBAPS and 24 to the non-reference genome group.

### Independent infections in a single patient

The patient sampled twice, in 2014 (14_01223) and 2017 (17_01604), had both isolates taken from a large lesion (diameter 80 mm) at two different locations on the lesion. The patient received treatment between the sampling times but continued to relapse (leishmaniasis recidivans). These two isolates were genetically different from each other with no evidence of a mixed infection (Fig 1): 14_01223 and 17_01604 had 159,535 genome-wide SNPs between them, including 1,616 on chromosome 2, and 15,606 on chromosome 36 (Fig 2).

14_01223 and 17_01604 were both allocated to the same non-reference genetic group, but when examined individually 14_01223's chromosomes 2 and 36 were genetically related to the reference group. 14_01223 had long runs of homozygosity (LROHs) across all of chromosome

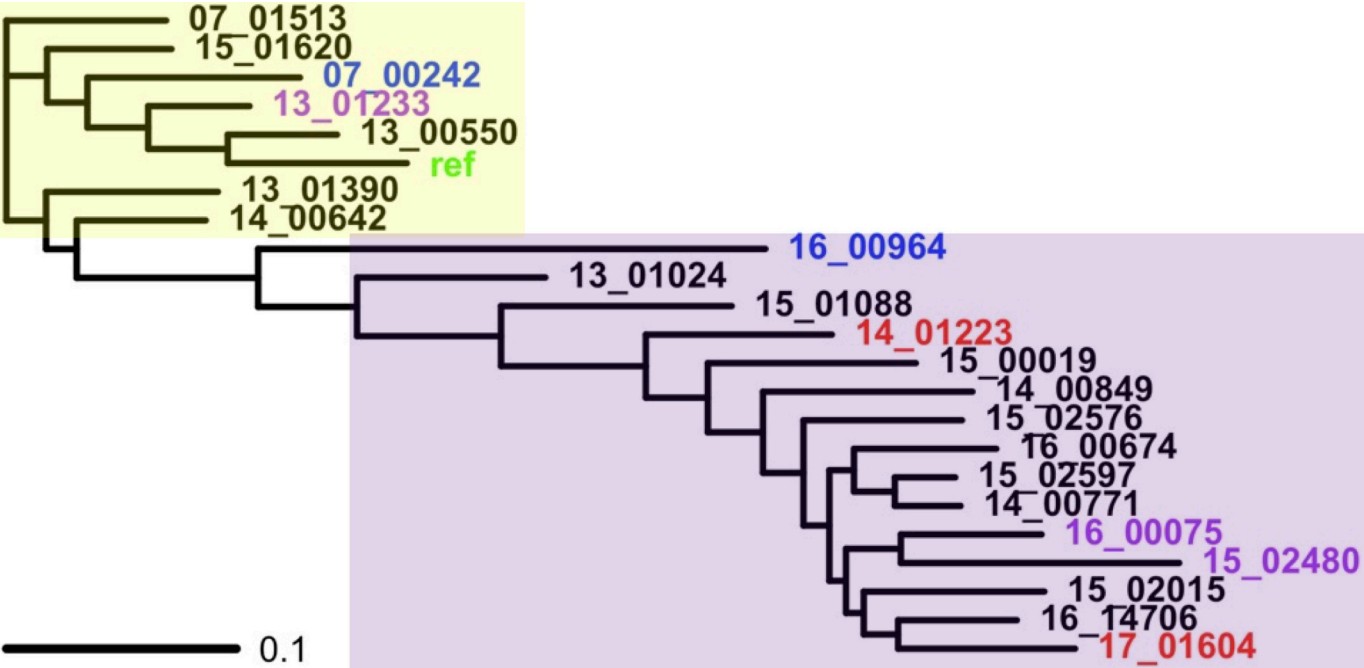

**Fig 1. A phylogeny constructed from all chromosomal SNPs showing the relatedness of the 22 isolates with the *L. tropica* reference genome ("ref", green).** The samples were from Syria (n = 17, black/red), Afghanistan (n = 3, purple) and Iran (n = 2, blue). Isolates 14_01223 and 17_01604 were from the same patient (red). The inferred genetically distinct groups from FastBAPS are represented by the *L. tropica* reference (yellow area) and non-reference groups (purple area).

2 and at chromosome 36 >1.63 Mb to the end at 2.715 Mb (Fig 2). In contrast, 17_01604 was heterozygous throughout. This difference in variation was confirmed by the SNPs' RDAF distributions which were skewed for 14_01223 but normal for 17_01604 (S3 and S4 Figs). The chromosome 36 of 14_01223 displayed heterozygosity at <1.63 Mb, then a LROH at 1.63–1.78 Mb, followed by near homozygosity >1.80 Mb. Based on inferred patterns from *L. major*, the 1.63 Mb breakpoint coincided with an acH3 mark, and the 1.78–1.80 Mb region corresponded with a putative transcription start site (TSS) near an inferred SSR. The patient had a mixed infection with two unique isolates, allocated to the non-reference genetic group, but with clear genomic differences.

## Varied ancestry and heterozygosity at chromosome 36

Further examination of chromosome 36 in 14_01223 with 14_00642, 16_00075, 16_00964 and 07_00242 presented sharp homozygosity-heterozygosity switches (Fig 3). 07_00242 and 16_00964 had similar patterns like a change at an inferred SSR and putative acH3 mark at 1.42 Mb, where the pattern of heterozygosity changed to a LROH until 1.8 Mb, before moving to homozygous similarity (Fig 2). 16_00075 had a jump in the RDAF at 1.28 Mb (at acH3 and baseJ marks) before a RDAF drop at 1.75 Mb that was also in 14_00642—neither of these samples had LROHs. All these five samples were disomic for this chromosome, except for trisomy in 14_00642 (S4 Fig).

The heterozygous and homozygous SNP densities, phylogenetic patterns, FastBAPS population assignment (S5 Fig), and mean RDAF consistent with somy levels at 0–1.28 Mb of chromosome 36 matched the genome-wide patterns and main trends in other chromosomes (Fig 3). The region at 1.28–1.60 Mb was heterogeneous due to the higher number of potential recombination breakpoints (Fig 3). At 1.60–1.78 Mb, 14_01223, 07_00242 and 16_00964 had

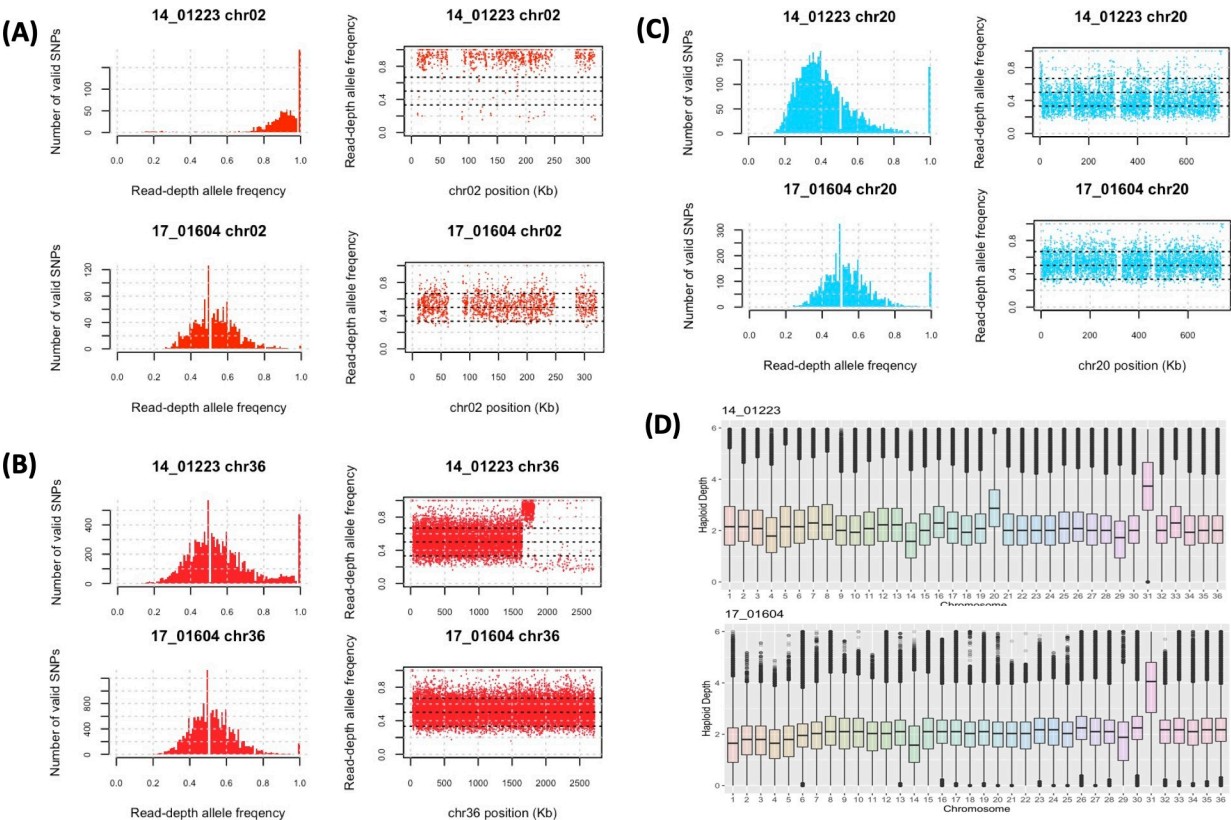

**Fig 2. 14_01223 and 17_01604 were from the same patient originally from Syria but were genetically distinct.** The genetic differences between these were illustrated by the SNP RDAF distributions (left of each panel) and the RDAF levels across the chromosome (right of each panel). (A) For chromosome 2, 14_01223 was nearly homozygous, whereas 17_01604 had heterozygosity comparable with most other samples (S3 Fig). (B) 14_01223 had a region with a high RDAF at 1.63–1.78 Mb followed by homozygosity at >1.80 Mb that 17_01604 did not have. (C) Chromosome 20 was approximately trisomic in 14_01223 but disomic in 17_01604 as shown by the RDAF distributions. (D) The haploid depth of each chromosome showed minimal differences except for chromosome 20.

LROHs, contrasting with the lower relative heterozygosity in the other 19 samples. Moreover, these three samples had unique ancestry (A) for this region, differentiating them from the expected reference (R) and non-reference (N) groups based on FastBAPS analysis (S5 Fig). This deviation in genetic ancestry continued for the rest of the chromosome, and all four strains were found to be related to the reference group, which was surprising for 07_00242, 16_00964 and 16_00075 had been assigned to the non-reference group at the whole-genome level. 16_00075 was unambiguously disomic for this chromosome, implying that the change in the main RDAF peak of ~0.5 at 0–1.28 Mb to a peak of ~0.75 at 1.28–1.75 Mb and then to ~0.25 at >1.75 Mb (S6 Fig) was a mix of cells with varied ancestries including a genetically different middle segment. Like all 22 samples here, 16_00075 was heterozygous at <1.28 Mb on this chromosome (R/N genotype). At >1.8 Mb, ~50% of 16_00075's cells were likely R/R, and ~50% were R/N, yielding a mean RDAF of ~0.25. In the middle at 1.28–1.75 Mb the other ancestry (A) implied a mosaic of R/A genotypes. The shift in pattern indicated a genetic exchange that may be the result of a sexual reproduction.

## Consistent recombination breakpoints at tetrasomic chromosome 31

The *Leishmania* chromosome 31 often shows a pattern of tetrasomy and high heterozygosity and this was also observed here (S4 Fig), indicative of likely homologous recombination across

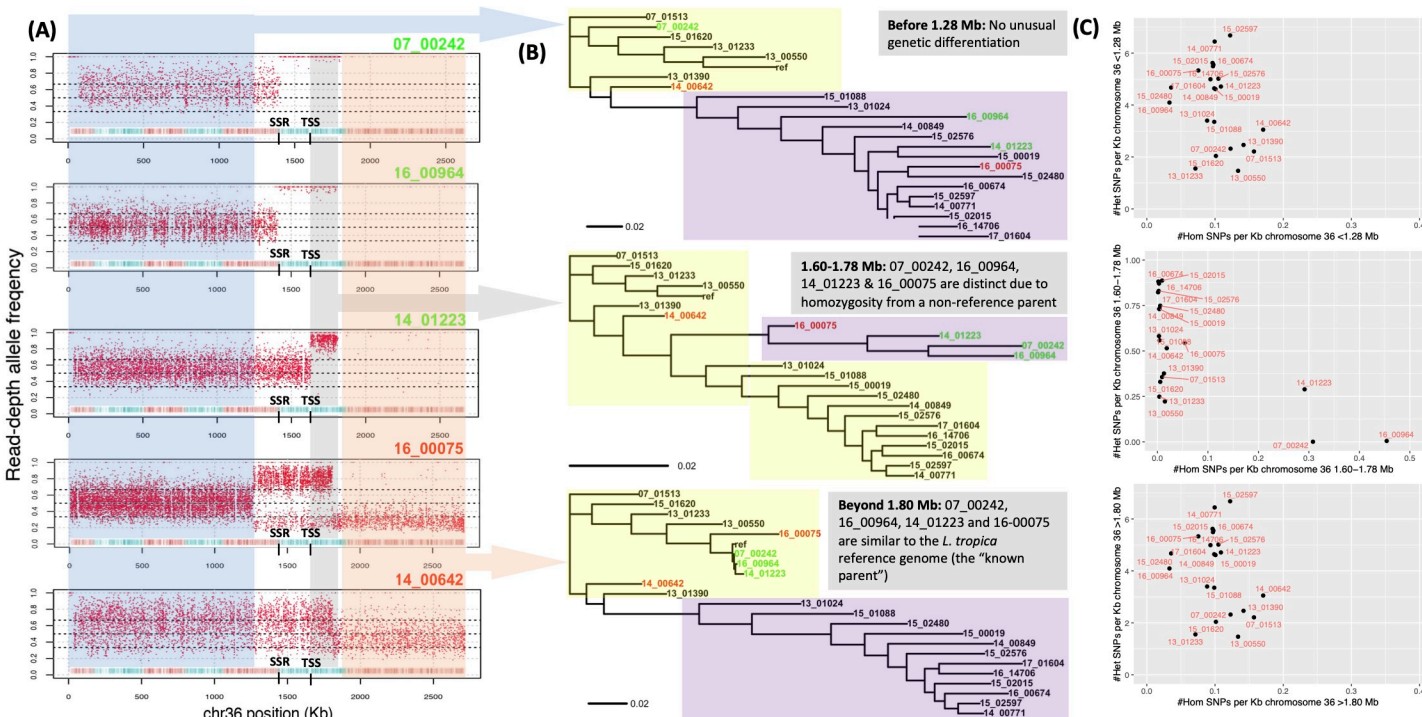

**Fig 3.** RDAF distributions (A) and phylogenies (B) constructed from SNPs at the 5'-end of chromosome 36 (<1280 Kb, blue RDAF, top phylogeny), middle (1600–1780 Kb, grey RDAF, middle phylogeny) and 3' end (>1800 Kb, orange RDAF, bottom phylogeny) showing the genetic relatedness of the 22 isolates with the *L. tropica* reference genome ("ref"). (A) Patterns of heterozygosity (blue), a LROH (grey) and homozygous similarity (orange) are seen in 14_01223, 07_00242 and 16_00964 (green), and high heterozygosity in 16_00075 and 14_642 (red). All isolates were disomic for chromosome 36, except 14_00642 that was trisomic. The putative TSS inferred from *L. major* is at 1.63 Mb (black), corresponding to a recombination breakpoint in 14_01223 (middle). 07_00242 (top) and 16_00964 (second) had breakpoint at an inferred strand-switch region (SSR) at 1.42 Mb. (B) The inferred genetically distinct groups are represented by the *L. tropica* reference (yellow area) and non-reference groups (purple area). (C) The homozygous (x-axis) and heterozygous (y-axis) SNPs/Kb for all 22 samples for the 0–1.28 Mb region (top), 1.60–1.78 region (middle) and >1.8 Mb region (bottom).

all chromosome copies and possible accelerated mutation rates [49]. Consistent with this faster chromosomal substitution rate, the haploid *L. tropica* reference had high divergence at this chromosome (average across 22 samples 8.19±2.15 SNPs/Kb) compared to the other 35 chromosomes (5.01±0.66 SNPs/Kb, t = -34.8, p = 4.4x10$^{-16}$). This was due to a high rate of heterozygous SNPs (average 8.13±2.17) relative to the other 35 chromosomes (5.85±-0.55 SNPs/Kb, t = -36.6, p = 4.4x10$^{-16}$), but this was not evident for homozygous SNPs (0.06±0.06 vs 0.15±0.17 SNPs/Kb). The RDAF distribution of chromosome 31 of heterozygous SNPs was second lowest of the chromosomes (S7 Fig), suggesting more recombination resulting in consistent heterozygosity. The FastBAPS genetic group allocation was consistent with differing heterozygous SNP rates between the reference and non-reference groups, potentially indicating a lack of genetic exchange between these groups for this chromosome (Fig 4).

Chromosome 31 had sharp changes in heterozygosity at 220 Kb and 950 Kb in samples 15_02015 and 15_01620, and at 950 Kb in 15_02597 (for which chromosome 31 was pentasomic) (Fig 4).

In 15_01620, the major RDAF switched from a mode of 0.75 at <220 Kb to 0.5 at 220–950 Kb, before increasing to 0.75 at >950 Kb (Fig 4). In 15_02576, the major RDAF modes were 0.4 and 0.6 <950 Kb, followed by 0.6 at >950 Kb (Fig 4). 15_02597's RDAF had a major peak of 0.5 <950 Kb and a jump to 0.8 at >950 Kb (Fig 4). In 16_00075, the major RDAF peak was between 0.5 and 0.6 throughout (Fig 4). 15_02015's RDAF peaks were at 0.25 and 0.5 <220

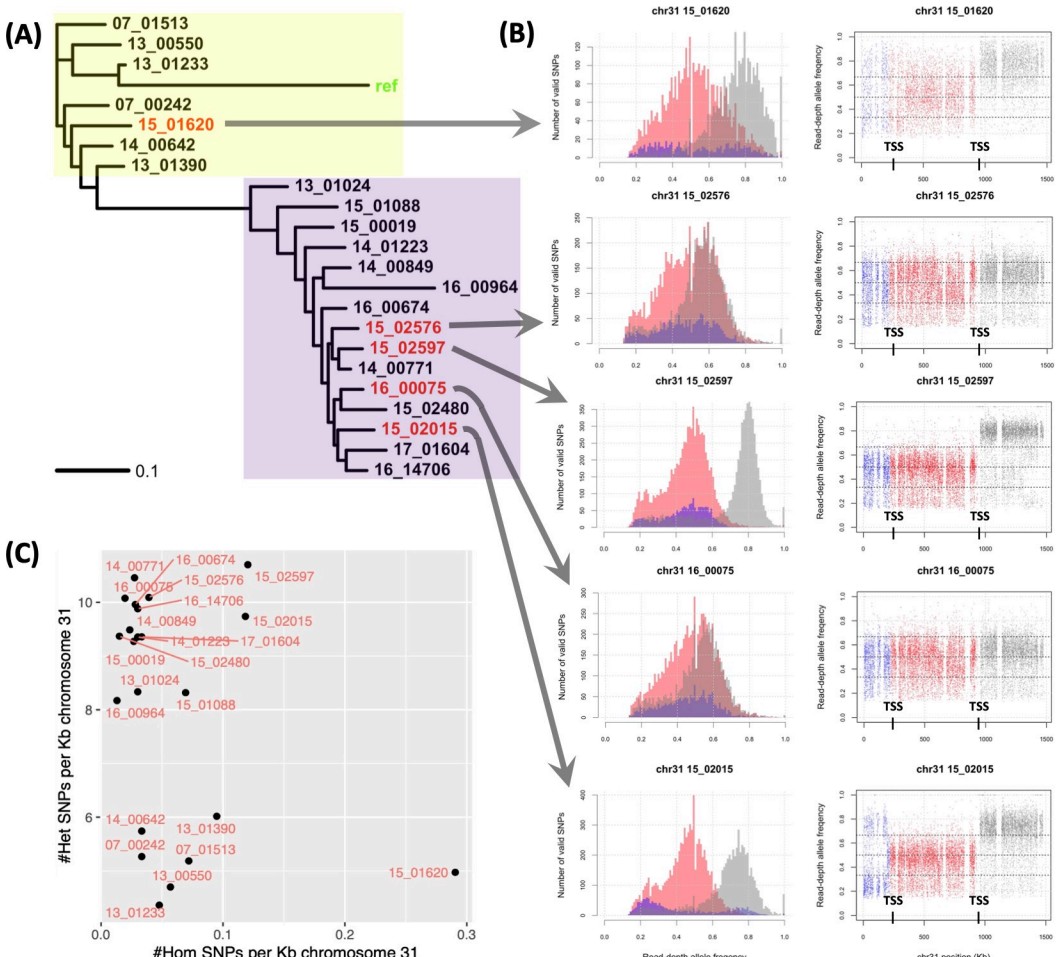

**Fig 4. Heterozygosity was high and haplotype-specific on chromosome 31.** (A) A rooted phylogeny constructed from SNPs at chromosome 31 showing a divergent *L. tropica* reference ("ref", green) due to a lack of heterozygous SNPs in this haploid genome. The inferred genetically distinct groups are represented by a group more related to the *L. tropica* reference (yellow area) and the non-reference group (purple area). (B) The read-depth allele frequency (RDAF) SNP densities (left) and distribution (right) across chromosome 31 (left) showed distinct patterns at the 5' end (<220 Kb, blue), middle (220–950 Kb, red) and 3' end (>950 Kb). 15_01620 (top) and 15_02015 (bottom) had recombination breakpoints at 220 and 950 Kb, coinciding with two putative TSSs inferred from experiments in *L. major* were at 220 and 950 Kb (black). 15_02597 (middle) also had a breakpoint at 950 Kb. 15_02576 (second) and 16_00075 (fourth) had no clear breakpoints. (C) The homozygous (x-axis) and heterozygous (y-axis) SNPs/Kb.

Kb, then 0.5 at 220–950 Kb, before 0.75 at >950 Kb (Fig 4). 220 Kb and 950 Kb are at TSSs containing acH3 marks, potentially matching TSSs inferred from experiments in *L. major*. These results demonstrate the high level of recombination at discrete breakpoints on this chromosome.

## A geographic basis for unique ancestry and variation at chromosome 2

Sample 16_00964 from Iran, which also had mixed heterozygosity with putative novel ancestry at chromosome 36, also showed the same pattern on chromosome 29 at >440 Kb, and this was shared with sample 07_00242 (at >490 Kb) from Iran as well. Both had heterozygosity up to 440 (16_00964) or 490 (07_00242) Kb as did the other 20 samples, followed by a LROH from this point to the end of the chromosome at 1,520 Kb (Fig 5). This 440–490 Kb region was near

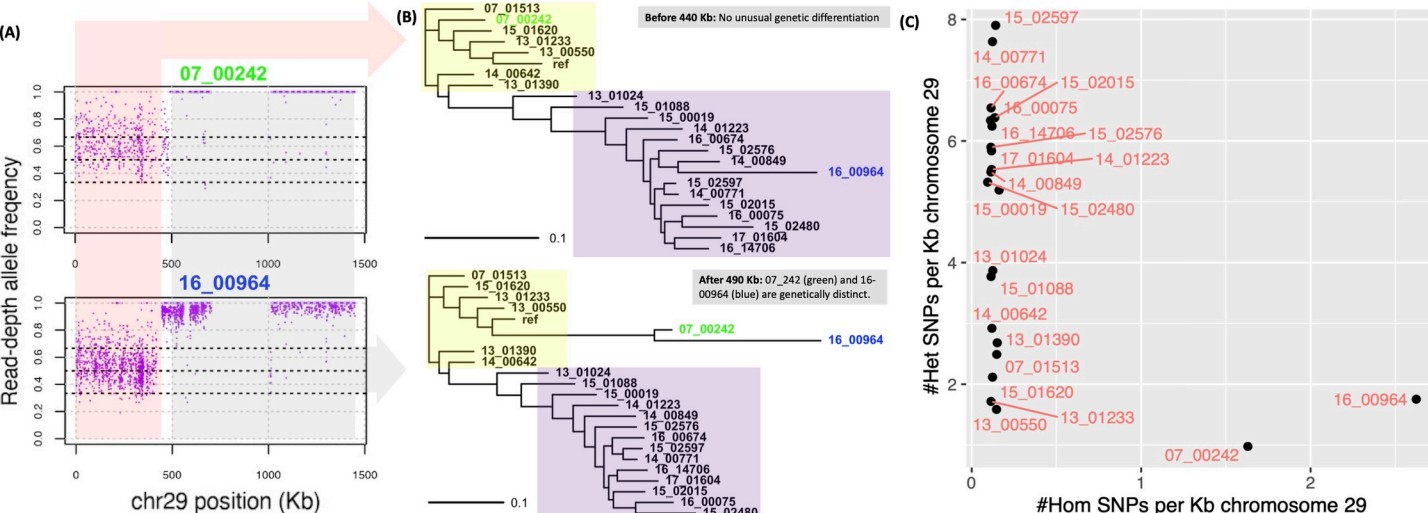

**Fig 5.** RDAF distributions (A) and phylogenies (B) constructed from SNPs at chromosome 29 from 0–440 Kb (pink RDAF, top phylogeny) and >490 Kb (grey RDAF, bottom phylogeny) showing the genetic relatedness of the 22 isolates with the *L. tropica* reference genome ("ref"). This showed unique segments of genetically distinct ancestries in 07_00242 (green) and 16_00964 (blue)—both came from patients originally from Iran. All samples were disomic for chromosome 29. The inferred genetically distinct groups are represented by the *L. tropica* reference (yellow area) and non-reference groups (purple area). (C) The homozygous (x-axis) and heterozygous (y-axis) SNPs per Kb for all 22 samples.

a TSS and a SSR. The genetic variation at >490 Kb in 16_00964 or 07_00242 was unlike the other samples and the reference genome, suggesting ancestry from an unsampled lineage (Fig 5). A similar picture of genetic exchange in another *L. donovani* lineage has been described near the region [40].

All three samples from Afghanistan (13_01233, 15_02480 and 16_00075) had 1.7-fold normalised average haploid read depth for chromosome 29 at 704–1,006 Kb (302 Kb total length), but the coverage was just 0.6 in the other 19 samples from Iran and Syria (S8 Fig). The rest of this chromosome was uniformly disomic in all samples (average depth 1.99±0.08), in line with the other chromosomes (average excluding chromosome 31, 2.12±0.33). This drop in read coverage was confirmed visually based on the read mapping distribution to the reference genome and again with the 13_00550 PacBio assembly (S9 Fig). This heterozygous deletion or contraction was present in the Iran- and Syria-linked samples from both the reference and non-reference genetic groups, and not the Afghanistan-linked samples, again from both reference (13_01233) and non-reference (15_02480 and 16_00075) groups (Fig 1).

## Geographic structure associated with heterozygosity loss on chromosome 23

Like chromosomes 29 and 36, the region <250 Kb at chromosome 23 had a segment of ancestry that was distinct from the reference and non-reference patterns with high heterozygosity in five samples (all from Iran or Afghanistan) that formed their own genetic group in the FastBAPS population assignment (S10 Fig). This difference in ancestry was due to the high level of homozygous differentiation from the reference genome of the other 17 samples from Syria at <250 Kb, unlike the high heterozygosity seen in 07_00242 and 16_00964 (both Iran), and 13_01233, 15_02480 and 16_00075 (all Afghanistan) (Fig 6). Chromosome 23 had a higher density of homozygous SNPs/Kb (1.02±0.56) compared to the other 35 chromosomes (0.13 ±0.07 SNPs/Kb, t = -74.6, p = 4.4x10$^{-16}$) and had the lowest mean RDAF using heterozygous SNPs (0.43±0.06 per sample) of all the chromosomes (average of the other 35, 0.52±0.04) (S7

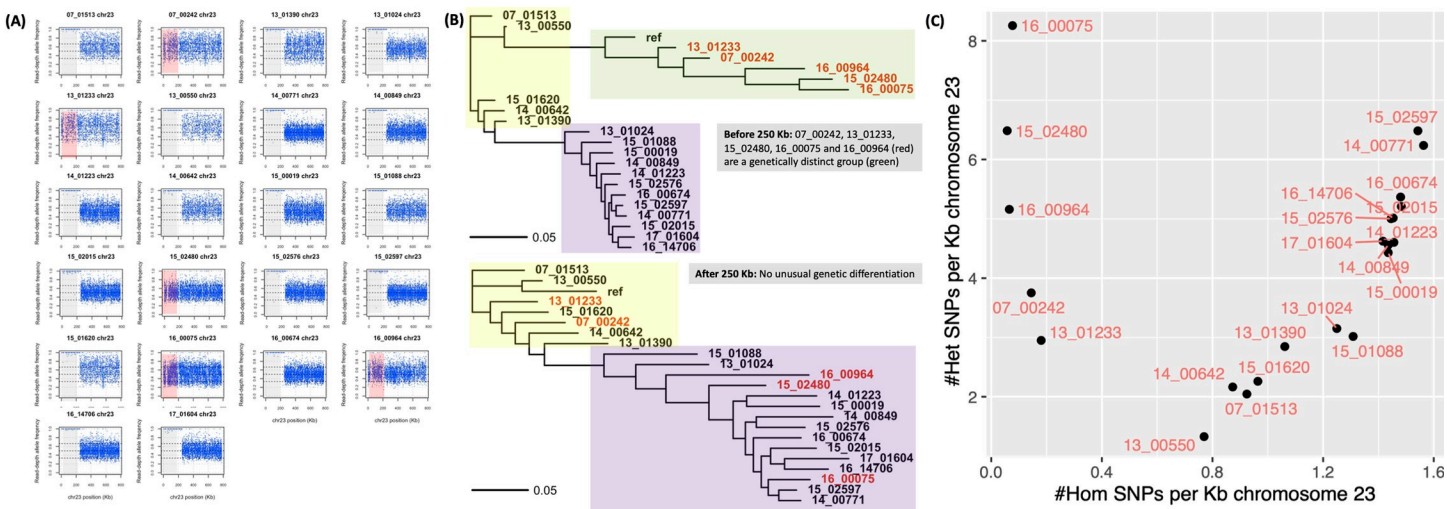

**Fig 6.** RDAF distributions (A) and phylogenies (B) constructed from SNPs at chromosome 23 at 0–250 Kb (grey and red in RDAF plots, top phylogeny) and >250 Kb (uncoloured RDAF, bottom phylogeny) showing the relatedness of the 22 isolates with the *L. tropica* reference genome ("ref"). This showed unique ancestries at chromosome 23 at 0–250 Kb with patterns related to the *L. tropica* reference in 07_00242, 13_01233, 15_02480, 16_00075 and 16_00964 (red in RDAF plots, red in phylogenies) and different patterns in the other 17 isolates. The inferred genetically distinct groups are represented by the *L. tropica* reference (yellow area) and non-reference groups (purple area). (C) The homozygous (x-axis) and heterozygous (y-axis) SNPs per Kb for all 22 samples.

Fig). This high SNP rate meant that this chromosome had 16.3% of all SNPs in each sample (795±343 vs 4,092±3,064 across the other 35 chromosomes). At >250 Kb, all samples conformed to their genome-wide pattern, such that 07_00242 and 13_01233 reverted to membership of the reference group, and 15_02480, 16_00075 and 16_00964 clustered with the non-reference group (Fig 6).

The near-uniform disomy of this chromosome in our isolates (average 2.15±0.27) and evidence of genetic drift from the high positive correlation of heterozygous and homozygous SNP densities in the 17 samples with LROHs suggested that recombination and somy changes were rare at this chromosome. We confirmed these results by read mapping the Ion Torrent library of 13_00550 to its own PacBio assembly to verify an absence of heterozygous SNPs before the 260 Kb position, followed by the heterozygosity >260Kb observed when mapping to the reference genome. This contrasted sharply with previous work on the *L. donovani* complex where chromosome 23 was typically trisomic with extensive diversity and encoded an amplification of the H-locus drug-resistance region [80].

## Monosomy and trisomy on chromosome 10

Chromosome 10 was disomic in all isolates except 14_00642, which was monosomic at 20–250 Kb and approximately trisomic at <20 and >250 Kb (Fig 7). These results based on the normalised haploid depth from read mapping to the *L. tropica* reference genome were replicated in mapping to the contigs of a *de novo* Ion Torrent assembly of 14_00642 (Fig 7), and again when mapping to the PacBio 13_00550 assembly, and these unusual patterns were confirmed visually in IGV (S11 Fig). 14_00642 had a much higher rate of homozygous SNPs at chromosome 10 than the other samples (S12 Fig). The sub-telomeric regions of this chromosome could be amplified, coinciding with potential SSRs at 20 Kb and 520 Kb based on *L. major* experiments (Fig 7). Its RDAF distribution had peaks at ~0.67 at >250 Kb and at ~0.33 at >520 Kb, consistent with trisomy or possibly a mixed cell population. The switch from mono- to tri-somy was near a putative SSR at ~270 Kb, 5' of the inferred centromere at ~295–301 Kb

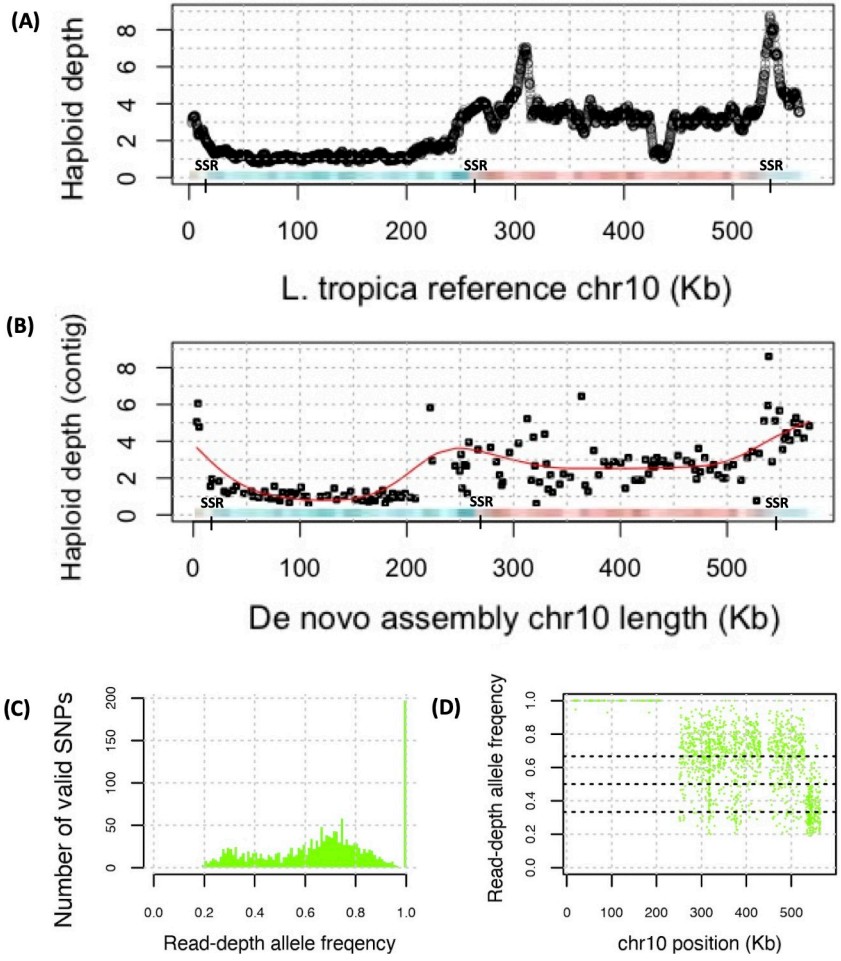

**Fig 7. The normalised haploid depth (top) from read mapping to the *L. tropica* reference genome and for the contigs from a *de novo* assembly of 14_00642 (middle).** Chromosome 10 potentially has three strand-switch regions (SSRs) inferred from experiments in *L. major* (red and blue panel). The lower panels show the numbers of valid SNPs (bottom left, green) and the RDAF for 14_00642.

[81] (Fig 7). 14_00642's pattern of low heterozygosity and LROHs was not in the other isolates except 15_02480, which had a LROH at >520 Kb, but had uniform disomy, like the other 20 samples (S12 Fig). The 230–250 Kb region contained numerous reference genome gaps and regions of low read mapping coverage, preventing clear delineation of a hypothetical break between the putative 5' chromosome 10.1 and 3' chromosome 10.2. Despite this, it is possible that this centromeric region delineates the formation of an acrocentric chromosome.

## Aneuploidy as a mechanism for more allelic variation, recombination and adaptive paths

We found 17 mutations at 11 SNPs that were triallelic, such that two derived alleles were present. These were in 14 of the 22 samples, which had between four and eleven such multi-allelic SNPs each. The somy level of chromosomes with triallelic SNPs (4.32±0.39) was higher than that of chromosomes with biallelic SNPs (2.11±0.33, p = 4.5x10^-7). In addition to these unusual LROHs at chromosomes 10, 23, 29, and 36 above, we observed rare switches to homozygosity for chromosomes 2 in 14_01223 (entire chromosome, S3 Fig), 4 in 07_00242 (entire

chromosome, S13 Fig), 11 in 07_01513 (at >380 Kb, S14 Fig), 12 in 15_02480 (at >330 Kb, S15 Fig), 13 in 07_00242 and 16_00964 (at >530 Kb, S16 Fig), 14 in 14_00642 (at <70 Kb, S17 Fig), 17 in 07_00242 (at <80 Kb, S18 Fig), 22 in 07_00242 (entire chromosome, S19 Fig), 24 in 07_01513 (at <150 Kb, S20 Fig), 27 in 13_01390 (at <1.05 Mb, S21 Fig), 28 in 14_00771 (at <180 Kb) and 15_02480 (at <210 Kb, S22 Fig), 30 in 15_02015 (at <120 Kb, S23 Fig) and 33 in 07_00242 (at >1.2 Mb, S24 Fig). Furthermore, the most common changes in heterozygosity frequency for chromosomes 10, 23, 31 and 36 driven by recombination and (in some cases) somy changes were akin to less frequent changes on chromosomes 8 (at <110 Kb and >380 Kb) in 13_01233, 14_00642, 15_01620 and 16_00075 (S25 Fig), 22 (at <90 Kb and >460 Kb) in 14_00642 (S19 Fig), 24 (at <90 Kb and 200 Kb) in 14_00771 (S20 Fig), 25 (different small sections) in 14_00771 and 15_02597 (S26 Fig) and 27 (at >1.05 Mb) in 13_01390 (S21 Fig). Moreover, LROHs similar to those in chromosomes 23, 29 and 36 described above, associated with potential regions of ancestry different from the reference and non-reference groups, was observed in chromosomes 28 in 16_00964 (at <740 Kb, S22 Fig), 32 in 07_00242 (entire chromosome) and in 15_02015 (at <1.0 Mb, S27 Fig). This pervasive recombination was underpinned by the absence of a deviation from the genome-wide patterns of group clustering in the kDNA SNP patterns (S28 Fig).

## Discussion

Here, we presented the genetic diversity and high degree of genomic variation within 22 isolates of *L. tropica* and found 301,659 SNPs across all samples, a number comparable to previous work [46]. We detected high heterozygosity and diverse LROHs, resulting in no clear correlation between the number of homozygous and heterozygous chromosomal SNPs per sample, in line with prior studies. This implied recombination and aneuploidy as key factors driving patchy genome-wide heterozygosity arising from older hybridisation events. In particular, chromosome 36 had shared patterns of heterozygosity followed by homozygous differentiation and then homozygous similarity, all on the same chromosome in 14_01223, 07_00242 and 16_00964. Our pattern of recombination breakpoints was similar to or exceeded rates of one cross-over per 1.5 Mb [32] or per 2.3 Mb [40], though an exact rate within *L. tropica* remains unclear. Across the genome, the extensive nature of the LROHs were indicative of regular recombination, and this pattern was present in both the main reference and non-reference genetic groups. These patterns suggested an increase somy allowed more recombination between chromosomes, a higher rate of derived allele accumulation, and hypothetically more diverse options following reversion to disomy associated with random loss of one chromosome copy.

Chromosome fission and fusion arise due to errors in chromosome replication and are rare in *Leishmania*. They tend to correlate with SSRs [82] that may be detected from histone H3 acetylation marks in *L. major* [55,83] and may preserve transcription of the PTUs. All except 14_00642's chromosome 10 were normal, and the 230–250 Kb region contained numerous reference genome gaps and regions of low read mapping coverage, preventing clear delineation of a hypothetical break between a putative 5' chromosome 10.1 and a 3' chromosome 10.2. Consequently, if a model of transient monosomy is excluded, then 50% of 14_00642's cells may have an intact disomic chromosome 10 and 50% may have a tetrasomic chromosome 10.2. Alternatively, all cells may have a disomic chromosome 10.2, and 50% may have an intact disomic chromosome 10 as well. Due to the effects of PCR during library preparation and the single-ended nature of these read libraries, they were insufficient to address subsequent questions on the location and sizes of the telomeric and centromeric DNA regions at new hypothetical chromosomes, though a variety of mechanisms exist to ensure cells can mitigate these chromosomal changes [84].

Telomeric amplification helps stabilise the genome during replication [85]. In *Leishmania*, the maintenance of telomeres occurs through rolling circle replication via extrachromosomal amplification, as in *Kluyveromyces lactis* [86,87] rather than telomeric loop formation [78]. And outside of the S cell cycle phase, DNA replication has been detected proximal to the chromosome telomeres, resembling a subtelomeric DNA synthesis activity [56]. Our isolates had extensive aneuploidy and higher heterozygosity than previously reported [46]. Recombination breakpoints coinciding with TSSs or SSRs were observed on numerous other chromosomes, which might affect DNA replication.

*L. tropica* likely has a higher capacity for interspecies mixing compared to other *Leishmania* and can create hybrids *in vivo* and *in vitro* [24,32,37,40,88]. *L. tropica* has a high efficiency of hybrid formation in co-infected sand flies and potentially greater capacity to generate mating-competent hybrids [32]. Genetic exchange occurs during reproduction by promastigotes in sand flies, and patterns of high and varied haplotype diversity were apparent in our results. This raises questions on the role of vector-species compatibility as a trigger for a sexual reproduction, and the potential for different sand flies to have varying microenvironments that differentially facilitate genetic exchange and hybrid generation. An important emerging question thus is the molecular basis of *L. tropica*'s broad vector compatibility, and how local environments mediate this [80,89].

Mosaic aneuploidy may originate through multiple mechanisms, not limited to meiotic nondisjunction, gene conversion [29], mitotic nondisjunction resulting in misallocating chromosomes [47], recombination-directed replication inferred from work on related parasites [55], and chromosomal replication followed by genome erosion [46]. To maintain the mosaic aneuploidy, Sterkers et al [30] argued for a parasexual reproduction, rather than a classical meiosis-like cycle, based on asymmetric chromosome allotments during mitosis. The distinctive heterozygosity blocks and aneuploidy among our isolates could be explained by either meiosis followed by selfing and chromosome copy number variation, or alternatively parasexual reproduction involving a tetraploid intermediate, or indeed both [88]. Recombination involving mobile genetic elements have also been suggested as a possible mechanism for the creation of mosaic aneuploidy [90].

Effect of long-term culturing of *Leishmania* and the environmental condition *in vitro* have previously been studied [91,92]. Structural variation, changes in ploidy [93] and altered gene dosage [47,94] has been observed in culture. Our isolates were cultured when diagnosed, before being stored at -156˚C and then re-cultured as promastigotes in axenic culture *in vitro* for two to six weeks before being DNA sequenced. Consequently, polymorphisms may have arisen during these steps and indeed some may have been lost. Moreover, although our study had the advantage of longer reads than typical Illumina runs, it lacked the information to resolve structural changes using paired read information and higher read depth would yield more precision in assessing copy number variation across chromosomes and subtle changes in heterozygosity. Notwithstanding our long read sequencing of one isolate, improving the quality of our reference genomes can spur new insights into regions with high structural variability and complex ancestry that may have been missed in this study.

## Conclusion

The high diversity, frequent changes in heterozygosity and abundant aneuploidy in these 22 *L. tropica* isolates were consistent with genetic exchange, possibly by sexual or parasexual mechanisms. Different frequencies of recombination have been observed between different *Leishmania* populations [80]. *L. tropica—L. donovani* hybrids that have spread to geographically distant regions though diverse *Phlebotomomus spp.* by genetic exchanges to adapt to the vector

have been described [16,28,89]. It remains unclear if this is due to intrinsic differences in *Leishmania* or due to differences in vector species. What is clear is that *L. tropica* has adapted to several vector species, which seems to increase hybridization, and can produce new progeny with genetically highly similar or extremely distinct *Leishmania*. The diverse ancestries, mixing and existence of hybrids may help explain the challenges in treating CL and leishmaniaisis revidans caused by *L. tropica* in the Middle East, but further research needs to be carried out to better understand the correlations between genetic diversity in *Leishmania*, the differences among the vectors and treatment outcome.

## Supporting information

**S1 Table. The number of total, valid, and discarded sequence reads per sample, along with the numbers of candidate and valid SNPs per sample.**
(TIF)

**S1 Fig.** The read coverage (A), mapping quality (MQ) (B), reverse reads (C) and forward reads (D) (all on x-axes) varied across the 22 samples (coloured dashed lines) compared to the number of SNPs called (y-axes).
(TIF)

**S2 Fig. The numbers of homozygous (x-axis) and heterozygous (y-axis) chromosomal SNPs per sample.**
(TIF)

**S3 Fig. 14_01223 had evidence of near-homozygosity on chromosome 2 where it had 539 homozygous SNPs (>6 times more than all the other samples) and only 359 heterozygous SNPs (far fewer than the others).**
(TIF)

**S4 Fig. The chromosomes' (x-axis) distributions of their normalised haploid read depth per base (y-axis) for all 22 isolates.**
(TIF)

**S5 Fig.** (A) Assignment of genetically distinct population using FastBAPS for SNPs at <1.28 Mb, 1.60–1.78 Mb, and > 1.80 Mb on chromosome 36 showing the genetic relatedness of the 22 isolates with the L. tropica reference genome ("ref", red group).
(TIF)

**S6 Fig. The read-depth allele (RDAF) distributions SNPs at chromosome 36's 5' end (<1280 Kb, red), middle (1600–1780 Kb, blue) and 3' end (>1800 Kb, grey).**
(TIF)

**S7 Fig. The read-depth allele frequency (RDAF) levels for heterozygous SNPs (y-axis) across chromosomes (x-axis) shown as boxplots highlighting the interquartile range.**
(TIF)

**S8 Fig. Chromosome 29 had a region at 704–1,006 Kb (302 Kb in length) with low depth (y-axis, normalised to haploid).**
(TIF)

**S9 Fig. Visualisation of the reads mapped to the reference genome at chromosome 29 at 678–732 Kb (top) and at 977–1,031 Kb (bottom) showing no change in coverage in 15_020480 (middle) compared to 14_00642 (top) with the heterozygous deletion.**
(TIF)

**S10 Fig. Assignment of genetically distinct population using FastBAPs for SNPs at chromosome 23.**
(TIF)

**S11 Fig. The read coverage and heterozygosity visualised for 14_00642's chromosome 10 reads for bases 0–340 Kb (top) and 240–580 Kb (bottom) using IGV.**
(TIF)

**S12 Fig.** The read depth in isolates 15_02480 (top) and 14_00642 (bottom) when reads were mapped to the L. tropica reference genome (A) and then to their own de novo genome assemblies (B).
(TIF)

**S13 Fig.** 07_00242 had homozygosity spanning the whole of chromosome 4 based on the read-depth allele frequency (RDAF) distribution (A) showing zero heterozygous SNPs and the RDAF across the chromosome showing minimal changes bar 720 homozygous SNPs.
(TIF)

**S14 Fig. 07_01513 had evidence of heterozygosity followed by homozygosity >380 Kb on chromosome 11 where it had 652 homozygous (>7 times more than all the other samples).**
(TIF)

**S15 Fig. 15_02480 alone had a recombination breakpoint separating a heterozygous region at 0–330 on chromosome 12 from a homozygous one at >300 Kb.**
(TIF)

**S16 Fig. 07_00242 and 16_00964 had evidence of recombination breakpoints separating a homozygous region at >530 Kb on chromosome 13 from heterozygous regions 5' of this.**
(TIF)

**S17 Fig. 14_00642 had evidence of homozygosity at <70 Kb on chromosome 14 where it had 213 homozygous SNPs across the chromosome (90% more all the other samples), followed by heterozygosity >70 Kb.**
(TIF)

**S18 Fig. 07_00242 had homozygosity at <80 Kb on chromosome 17 following by heterozygosity.**
(TIF)

**S19 Fig. 07_00242 had homozygosity for all of chromosome 22.**
(TIF)

**S20 Fig.** 07_01513 (A) and 14_00771 (B) had evidence of recombination breakpoints separating a homozygous region at <90 Kb in 14_00771 and <150 Kb in 07_01513 on chromosome 24 from heterozygous regions 3' of this.
(TIF)

**S21 Fig. 13_01390 had evidence of quasi-homozygosity on chromosome 27 where it had 1,626 homozygous (>8 times more than all the other samples).**
(TIF)

**S22 Fig.** 15_02480 (A), 14_00771 (B) and 16_00964 (C) had evidence of recombination breakpoints separating a homozygous region at <180 Kb in 14_00771, at <210 Kb in 15_02480, and at <740 Kb in 16–00964 on chromosome 28 from heterozygous regions 3' of this.
(TIF)

**S23 Fig. 15_02015 had evidence of a homozygous region at <120 Kb on chromosome 30 where it had 479 homozygous (more than twice any other sample).**
(TIF)

**S24 Fig. 07_00242 had heterozygosity at <1.2 Mb on chromosome 33 following by mainly homozygosity >1.2 Mb.**
(TIF)

**S25 Fig. There was evidence of recombination breakpoints separating regions with a lower average read-depth allele frequency (RDAF) at 110–380 Kb from one with higher average RDAF at <110 Kb and >380 Kb on chromosome 8.**
(TIF)

**S26 Fig.** 15_02597 (A) and 14_00771 (B) and had numerous recombination breakpoints separating short regions of homozygosity on chromosome 25 from heterozygous regions.
(TIF)

**S27 Fig.** 15_02015 (A) 07_00242 (B) and had regions of homozygosity on chromosome 32 spanning the whole chromosome for 07_00242 based on the read-depth allele frequency (RDAF) distribution (left) and the dearth of heterozygous SNPs (right).
(TIF)

**S28 Fig. A co-phylogeny constructed from kDNA (left) and genome-wide (right) SNPs showing the relatedness of the 22 isolates.**
(TIF)

## Acknowledgments

The authors thank Hideo Imamura at the Universitair Ziekenhuis Brussel (Brussels, Belgium) for advice on bioinformatics protocols. The authors thank James A. Cotton at the Parasite Genomics Group, Wellcome Sanger Institute (Hinxton, United Kingdom) for access to the *Leishmania tropica* reference genome. The authors thank Karin Tegmark Wisell, Elisabeth Hallin-Bergvall, Lisbeth Gregory and Georgina Isak at the Public Health Agency of Sweden (Sweden) for contributing to the study.

## Author Contributions

**Conceptualization:** Hedvig Glans, Maria Bradley, Björn Andersson, Tim Downing.

**Data curation:** Hedvig Glans.

**Investigation:** Hedvig Glans.

**Methodology:** Hedvig Glans, Erik Alm, Tim Downing.

**Project administration:** Hedvig Glans, Björn Andersson.

**Resources:** Hedvig Glans, Maria Lind Karlberg, Reza Advani, Tim Downing.

**Software:** Erik Alm, Tim Downing.

**Supervision:** Hedvig Glans, Maria Bradley, Björn Andersson, Tim Downing.

**Visualization:** Tim Downing.

**Writing – original draft:** Hedvig Glans, Björn Andersson, Tim Downing.

**Writing – review & editing:** Hedvig Glans, Maria Lind Karlberg, Reza Advani, Maria Bradley, Erik Alm, Björn Andersson, Tim Downing.

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
