## [Decision Letter · Decision Letter 0]

19 Oct 2021

Dear Dr Glans,

Thank you very much for submitting your manuscript "High Genome Plasticity and Frequent Genetic Exchange in Leishmania tropica Isolates from Afghanistan, Iran and Syria" for consideration at PLOS Neglected Tropical Diseases. As with all papers reviewed by the journal, your manuscript was reviewed by members of the editorial board and by several independent reviewers. In light of the reviews (below this email), we would like to invite the resubmission of a significantly-revised version that takes into account the reviewers' comments. 

The authors are recommended to carefully address the reviewers' comments and to make available their data to the public as recommended by reviewer 2.

We cannot make any decision about publication until we have seen the revised manuscript and your response to the reviewers' comments. Your revised manuscript is also likely to be sent to reviewers for further evaluation.

Sincerely,

Ikram Guizani

Associate Editor

Brian Weiss

Deputy Editor

The authors are recommended to carefully address the reviewers' comments and to make available their data to the public as recommended by reviewer 2.

Reviewer's Responses to Questions

**Key Review Criteria Required for Acceptance?**

**Methods**

-Are the objectives of the study clearly articulated with a clear testable hypothesis stated?

-Is the study design appropriate to address the stated objectives?

-Is the population clearly described and appropriate for the hypothesis being tested?

-Is the sample size sufficient to ensure adequate power to address the hypothesis being tested?

-Were correct statistical analysis used to support conclusions?

-Are there concerns about ethical or regulatory requirements being met?

Reviewer #1: (No Response)

Reviewer #2: (No Response)

Reviewer #3: The authors have used whole genome sequencing for studying population genetics based on chromosomal ploidy and patterns of single nucleotide polymorphisms among a series of 22 L.tropica natural isolates, which is a sound approach.

Reviewer #4: The objectives have not been defined. Lines 142-147 and the entire Methods section refer to the isolation and whole genome sequencing and analysis of 22 Leishmania tropica isolates from patients with the aim of finding genetic diversity and group them in the context of their geographical origin. This design is not appropriate to test the existence of meiosis and parasexual processes in the parasite. No experiment has been planned to test these hypotheses (see comments below).

Ethical approval was received according to the ethics statement in the manuscript, although I don't have access to the document.

Statistical analysis other than statistical checks in whole genome sequence analysis is not applicable here because only chromosomal sequence patterns are being addressed.

**Results**

-Does the analysis presented match the analysis plan?

-Are the results clearly and completely presented?

-Are the figures (Tables, Images) of sufficient quality for clarity?

Reviewer #1: (No Response)

Reviewer #2: (No Response)

Reviewer #3: The authors are transparent with their data as shown by the high number of supplementary files which are of sufficient quality. The quality of the original figures is likely not enough for publication (they appear fuzzy).

Reviewer #4: The results are well presented, match the analysis plan and show some examples of recombination and patchy heterozigosity, aneuploidy and structural variation. However, the experiment is not well planned to address the existence of meoisis and parasexuality in Leishmania (see comments below). 

Some panels in figures are too small to be read (axis titles and scales, for example).

**Conclusions**

-Are the conclusions supported by the data presented?

-Are the limitations of analysis clearly described?

-Do the authors discuss how these data can be helpful to advance our understanding of the topic under study?

-Is public health relevance addressed?

Reviewer #1: (No Response)

Reviewer #2: (No Response)

Reviewer #3: The conclusion of the paper ressembles more of an introduction than a conclusion. Their data is interesting and support genetic exchange. However it do not explain the adaptation of Ltropica to different vector species nor do it explain the challenges in treating CL by Ltropica. No such link with their data was made.

Reviewer #4: The conclusions are speculative. The patchy heterozygosity patterns found just suggest but not definitely support the existence of meiotic and parasexual processes. The results point to important recombination processes, but other recombination mechanisms based on mobile genetic elements that may explain such events have been described. For example, those based on the Pr77 hallmark. Also, aneuploidy has been associated to chromosome amplification as a mechanism of gene expression modulation. For example, it has been described in antimony-resistant strains. In fact, all patients included in the study to obtain the isolates were treated with different outcomes (cure, relapse or failure). Each clinical situation and patient-parasite interaction could have resulted in different amplification patterns or recombination processes under the selective pressure. The reduced number of samples makes addressisng these issues difficult. Therefore, the suggestions presented as conclusions have not been experimentally proven. At this preliminary stage, the results do not reach sufficient public health relevance.

**Editorial and Data Presentation Modifications?**

Reviewer #1: (No Response)

Reviewer #2: (No Response)

Reviewer #3: 1- There is a problem with Figure S4: the 3rd and 4th rows are the same (3rd row is repeated in the 4th row) and some strains are missing from the figure; this should be corrected.

2- Line 29: 'where IT is known...'

3- Line 33: 'tropica' shoul be in italic

4- Lines 57 and 61: 'species' should not be in italic

5- Line 73: change to 'Phlembotmine' to 'Phlebotomine' 

6- Lines 74 and 76: change 'serengeti' to 'sergenti'

7- Line 75: The beginning of the sentence up to 'zoonically' is not clear, should be rephrased.

8- Line 154: 'fluconazole'

9- Line 157: 'second sample IN 2017'

10- Line 179: 'at 23oC ' instead of in 23oC

11- Line 184: please define 'MGB'

12: Line177: no need to mention the samples that were discarded as these are not further discussed nor is an explanation provided for the low quality of their libraries.

13- Line 197: There is a lot of variability in the percentage of discarded reads between libraries and the authors should briefly discuss about the possible reasons explaning this difference in sequencing quality.

14- Line 210: from FigS1B their is not benefit of using MQ30 on the number of called SNPs. MQ55 or MQ60 looks more appropriate. Please described why MQ30 was used and how it was selected as a cutoff value.

15- Line 269-270: a parenthesis is opened at line 269 but is not closed (should be closed after ref 76)

16- Lines 274-276: It is not clear what these SNPs are. It is mentionned that 1594 SNPs were detected among the 22 samples but the next sentence says that '220 SNPs were found within our collection'. This seems inconsistent, please clarify.

17- Line 284: at line 190 the author mention >30-fold coverage. Please be consistent and correct for the proper coverage value.

18- Line 288: define 'SVs'

19- What do the authors mean by 'artefacts associated with the sequence quality'; on which criteria?

20- Line 308: for clarity please please specify the reference genome. I know it is mentionned in the Methods but this will help the reader appreciate what was done if repeated here (especially that later in the same paragraph the authors mention an alignement on a distinct reference from the assembly of 13_00550).

21- Line 315: 'sequencing OF isolate'

22- Line 316- 'like the other 21 isolates'. It is not clear to the reviewer whether the 21 isolates were also mapped to the 13_00550 assembly or if the authors are referring to the number of heterozygous SNPs detected from the alignment of the 21 isolates against the L590 genome.

23- Line 370: 'that may possibly have affected the treatment process'. This is highly speculative. Are there SNPs that would support such claim? In genes reputed to be implicated in treatment failure? or whose function could possibly be linked with drug response?

24- Lines 387-388: this is data interpretation and should not be part of the figure caption, which is about describing the figure and not discussing on its data.

25- Line 426: From FigS7 chr 23 seems to be the one with the lower RDAF, not chr 31.

26- Line 494: Shouln't it be Fig 6 instead of Fig 5?

27- Line 496: From FigS7, chr23 has the lowest RDAF not the highest.

28- Line 527: 13_0050 instead of 13_550

29- Lines 553-558: 'such as both derived alleles were present' what does this mean exactly? The same for '...the most interesting LROHs...were parralleled by more rare...' what do they authors mean by parralleled? 

30- Line 593: What does 'All bar' means?

31- Line 599: Here is a good place to discuss about the sequencing quality I mentionned earlier. What does the data miss to be able to address these questions?

Reviewer #4: Clarity in some figure panels should be improved.

**Summary and General Comments**

Reviewer #1: This well written paper provides a sound whole genome sequencing analysis of 22 strains of Leishmania tropica. The main conclusions drawn regarding genomic diversity of the strains that may reflect genetic exchange by sexual or parasexual processes are supported by the data and are consistent with prior studies on L. tropica and other Old World strains. While the findings may not represent a major advance, the report will nonetheless provide the field with informative new datasets that have been carefully analyzed and displayed by the application of currently available bioinformatic tools. 

There are a few minor points that can be addressed:

Lines 80-81: …..subsequently as amastigotes within mammalian monocytes (23). 

NOT correct

Lines 84-85: Evidence of meiosis and hybridisation has been observed in diverse natural isolates of Leishmania (25- 28 )

These studies provide evidence for hybridization, not meiosis. The mode of genetic exchange could not be inferred from these studies.

Lines 85-86: Reproduction in Leishmania is facilitated by its genes for meiosis prometaphase 1 (24) to allow crossing-over, resulting in recombinant chromosomes (30). 

There is as yet no evidence that any of the meiotic gene homologues in Leishmania are functional or play a role in genetic exchange. 

Legend to fig 2b and c are incorrect, need to be interchanged. 

The description of the two genetically distinct strains isolated from the same patient is very interesting. Since the genomes remain highly homologous, Is it possible that instead of a mixed infection, the genomic differences observed reflect selfing between the same strain that occurred in the vector, with simultaneous transmission of selfing and non-selfing promastigotes? Since genetic exchange involving intracellular amastigotes has recently been described (Telittchenko and Descoteaux, 2020), it is also possible that a selfing event occurred in ther

human host. 

Line 632: Effect of long-term culturing of Leishmania and the environmental condition in vitro have previously been studied (ref). 

Reference needs to be included

Reviewer #2: General comments

The authors report here a genome-wide SNP analysis of 22 isolates of Leishmania tropica, revealing extensive genomic variation and a very complex population structure with evidence of genetic exchange. While the results are consistent with previous observations, clear examples of different ancestries on same chromosomes is shown through a careful dissection of changes in heterozygosity.

While the study is limited in scope, not allowing the authors to examine genetic variation and phylogenies in the context of important metadata such as treatment outcome, it is carried out soundly and with robust methodology and data analysis.

Minor comments/corrections

Ln 111 – The wording suggests that Leishmania regulates gene expression through gene dosage (chromosome and gene copy variation) This may be a little misleading considering the significant amount of post-transcriptional regulation

Ln 182 – replace ‘were’ with ‘was’

Ln 186 – Bioproject PRJEB45563 is not public and there are no sequences associated with it. Data should be released before publication.

Fig 2. Labels for panels B and C are switched.

Ln 370 – ‘possible’ should be ‘possibly’

Reviewer #3: 1- Line 197: There is a lot of variability in the percentage of discarded reads between libraries and the authors should briefly discuss about the possible reasons explaning this difference in sequencing quality.

2- Line 210: from FigS1B their is not benefit of using MQ30 on the number of called SNPs. MQ55 or MQ60 looks more appropriate. Please described why MQ30 was used and how it was selected as a cutoff value.

3- Lines 213 and 306: 301659 SNPs is lower than the sum of the Candidate_SNPs column in TableS1. I guess this is because SNPs shared by different samples are counted as one SNP. If so please be mention this explicitely (either at line 213 or 306) and it would be nice to have a global idea of the level of SNPs redundancy between samples.

4- Line 222: What are these outlying chromosomes with extreme SNPs rates?

5- Line 313: Samples 16_00964 and 07_00242 seems to bring a lot of weight in the negative correlation. If removing these outliers there seems to be 2 groups having roughly the same number of homozygous SNPs but differenciated by the number of heterozygous SNPs. Is the correlation still holds when excluding the 2 outliers?

6- Lines 325-328: Why is sample 16_00964 labelled in Figure 1 as part of the non-ref group if about half of its chromosomes ambiguous? What is the statistics supporting its assignement to this group?

7- Line 344: the authors should explicitly mention the number of shared SNPs between samples 14_01223 and 17_01604. For now only the total number of SNPs in each of these samples is indicated. Also how much of the non-shared SNPs part of chromosomes 2 and 36? This would help the reader appreciate how much the heterozygosity of 17_01604 for these 2 chromosomes accounts for in the difference between these 2 samples isolated from the same patient.

8- The results section is sometimes confusing because some data is repeatedly described. For example the heterozygosity of chr36 in 14_01223 mentionned at lines 365-366 had already been mentionned at line 361-362 and it at first confusing whether this is a new piece of information of not. Another example is the sentence at lines 411-413 which is a repeat on the zygocity pattern of 16_00075 which had been described in the sentence just before (lines 408-410). Another example is the paragraph from lines 445 to 454. the authors start by describing RDAF profles for chr 31 in 4 samples, mentionningthe positions affected and the ID of the samples. then they come back after with more details for each samples by mentionning again the same positions. This is a lot of nucleotide IDs and RDAF for the reader to process. The authors should try to improve how they are conveying the data and associated interpretation throughout the Results section. For example for the later, it would be simpler to start the paragraph by saying that changes in heterozygocity were observed for samples A, B, C and D. then continue with the specific RDAF values and chr positions for each samples, and finish by indicating which of these positions are associated with TSS or other genomic features.

9- Line 380: The RDAF in Fig3 fits with chr36 diploidy for 16_00964 and 14_01223 but not for the others. How do the authors reconcile this discrepancy? The RDAF for chr36 in 07_00242 is indeed along the 0.6 line, suggesting putative triploidy. This is even clearer for 14_00642 with RDAF signals linning close to the 0.6 and 0.3 until position ~1800 and then with only the RDAF signal at 0.3 remaining. This kind of signals are highy suggestive of triploidy. For 16_00075 the RDAF data suggests diploidy until position ~1250, and then tetraploidy from ~1250 to ~1800 with RDAF signals at 0.75 and 0.25. 

The authors explain this for 16_00075 at line 410 by saying this is likely the result of 2 populations. This is possible, but not if cells were cloned prior to sequencing. This is something that should be mentionned in the Methods, i.e. were cells cloned prior to genomic DNA extraction or was genomic DNA extracted from the population. Also for how long were the cells passaged prior to genomic DNA extraction. It is mentionned in the Methods that 'isolates were grown for the minimum time necessary to produce adequate parasite numbers to isolate enough genomic DNA'. However the authors need to be more precise as passaging could have an influence here, for example by altering ploidy is cells were passaged for a long time (ploidy is known to randomly vary with time, at least for some chromosomes). 

10- Line 517: It would be interesting that in the discussion the author discuss on the ploidy of chr23 in their samples (mostly diploid) compared to other Leishmania samples or species (Ltrop or others). For example in Franssen et al eLife 2020, chr23 in Ldonovani isolates is the chromosome with the highest standard deviation in ploidy value, with most isolates being triploid. Why does this uniform diploidy would be specific to Ltropica?

11- Line 530: With next generation sequencing, it is not uncommon to observed putative amplification signal at sub-telomeric ends that are in fact artefactual. This would need to be validated, by Southern blots of digested genomic DNA followed by hybridization with a probe for a gene part of the amplicon and for a gene outside of the amplicon, to compare hybridization signal intensities (normalised by the hybridization signal for a gene not part of chr10). 

12- Lines 531-532: This only supports triploidy is the authors sequenced clones, hence the need adding this info to the Methods, as mentionned above. If they sequenced the population of parasites isolated from the lesion one cannot exclude that this is due to a mixed population, as suggested at line 410 for chr 36.

Reviewer #4: In this manuscript, a whole genome sequence strategy has been used to map SNPs in 22 Leishmania tropica isolates from treated patients in Sweeden with different outcomes .These patients were infected in the Middle East in a decade (2007-2017). According to this analysis, blocks of heterozygosity followed by homozygous blocks with breakpoints at strand-switch regions have been found. The authors conclude that these findings sugggest the existence of genetic exchange in possible parasexual processes, but this requires further experimental testing. In fact, chromosome amplification as a gene expression regulation mechanism, and recombination events through mobile genetic elements, may also explain the results.Therefore, experiments to determine which mechanisms are involved in the variations found should be conducted. No experiments have been performed to suggests the existence of meiotic processes in Leishmania. 

Language editing. The manuscript is well written but full of imprecise or not standard terms that should be defined. For example, homozygous differentiation, homozygous similarity or mixed heterozygosity.

PLOS authors have the option to publish the peer review history of their article (what does this mean?). If published, this will include your full peer review and any attached files.

Reviewer #1: No

Reviewer #2: No

Reviewer #3: No

Reviewer #4: No
---

## [Decision Letter · Decision Letter 1]

1 Dec 2021

Dear Dr Glans,

Thank you very much for submitting your manuscript "High Genome Plasticity and Frequent Genetic Exchange in Leishmania tropica Isolates from Afghanistan, Iran and Syria" for consideration at PLOS Neglected Tropical Diseases. As with all papers reviewed by the journal, your manuscript was reviewed by members of the editorial board and by several independent reviewers. The reviewers appreciated the attention to an important topic. Based on the reviews, we are likely to accept this manuscript for publication, providing that you modify the manuscript according to the review recommendations. 

The authors sequenced 22 L. tropica isolates from migrant patients originating from Syria, Iran and Afghanistan with the objective to study genetic diversity of these parasites at the isolate levels and between different isolates. 

The study highlights genomic diversity in these parasites and infers genetic mechanisms underlying the observed diversity. The study does not bring major advances but confirms previous observations obtained with other strains and brings precious resources to the scientific community. As such the links to zenodo should be included to the text in addition to the link to figshare.

The authors should also consider the comments about mobile genetic elements.

Sincerely,

Ikram Guizani

Associate Editor

Brian Weiss

Deputy Editor

The authors sequenced 22 L. tropica isolates from migrant patients originating from Syria, Iran and Afghanistan with the objective to study genetic diversity of these parasites at the isolate levels and between different isolates. 

The study highlights genomic diversity in these parasites and infers genetic mechanisms underlying the observed diversity. The study does not bring major advances but confirms previous observations obtained with other strains and brings precious resources to the scientific community. As such the links to zenodo should be included to the text in addition to the link to figshare.

The authors should also consider the comments about mobile genetic elements.

Reviewer's Responses to Questions

**Key Review Criteria Required for Acceptance?**

**Methods**

-Are the objectives of the study clearly articulated with a clear testable hypothesis stated?

-Is the study design appropriate to address the stated objectives?

-Is the population clearly described and appropriate for the hypothesis being tested?

-Is the sample size sufficient to ensure adequate power to address the hypothesis being tested?

-Were correct statistical analysis used to support conclusions?

-Are there concerns about ethical or regulatory requirements being met?

Reviewer #1: yes

Reviewer #3: (No Response)

Reviewer #4: I am asking about the approval document from the ethical board, not the patients' consent.

**Results**

-Does the analysis presented match the analysis plan?

-Are the results clearly and completely presented?

-Are the figures (Tables, Images) of sufficient quality for clarity?

Reviewer #1: yes

Reviewer #3: (No Response)

Reviewer #4: This has been adequately addressed.

**Conclusions**

-Are the conclusions supported by the data presented?

-Are the limitations of analysis clearly described?

-Do the authors discuss how these data can be helpful to advance our understanding of the topic under study?

-Is public health relevance addressed?

Reviewer #1: yes

Reviewer #3: (No Response)

Reviewer #4: The authors' haven't addressed the following issues:

"...all patients included in the study to obtain the isolates were treated with different outcomes (cure, relapse or failure). Each clinical situation and patient-parasite interaction could have resulted in different amplification patterns or recombination processes under the selective pressure. The reduced number of samples makes addressing these issues difficult. "

"At this preliminary stage, the results do not reach sufficient public health relevance". 

Even when the authors have toned down the conclusions, they are merely based in suggestions. Although the topic is important, I find that the work is unfinished and the manuscript inconclusive.

This highlights the importance of properly designing experiments before conducting them.

It would be convenient to include a reference for the new sentence about the mobile genetic elements. As the authors state in the response to the reviewers document, mobile elements are much less abundant in Leishmania spp. than in T. cruzi, but there is an exception: L. braziliensis and other species included in the Viannia subgenus.

**Editorial and Data Presentation Modifications?**

Reviewer #1: no comment

Reviewer #3: (No Response)

Reviewer #4: This has been adequately addressed.

**Summary and General Comments**

Reviewer #1: The revised manuscript has adequately addressed my main concerns and those of the other reviewers, except for rev #4. I am satisfied that while the paper does not provide a major advance, it does provide a valuable resource to the Leishmania population genetics community. While further studies as requested by rev#4 would indeed be required to support the mode of genetic exchange in these protists, I believe that they are beyond the scope of this paper.

Reviewer #3: (No Response)

Reviewer #4: The work is unfinished and the manuscript inconclusive (see my comments concerning the conclusions).

PLOS authors have the option to publish the peer review history of their article (what does this mean?). If published, this will include your full peer review and any attached files.

Reviewer #1: Yes: David Sacks

Reviewer #3: No

Reviewer #4: No

Figure Files:

Data Requirements:

Reproducibility:

References

---

## [Editor Report · Decision Letter 2]

17 Dec 2021

Dear Dr Glans,

We are pleased to inform you that your manuscript 'High Genome Plasticity and Frequent Genetic Exchange in Leishmania tropica Isolates from Afghanistan, Iran and Syria' has been provisionally accepted for publication in PLOS Neglected Tropical Diseases.

Best regards,

Ikram Guizani

Associate Editor

Brian Weiss

Deputy Editor

---

## [Editor Report · Acceptance letter]

28 Dec 2021

Dear Dr Glans,

We are delighted to inform you that your manuscript, "High Genome Plasticity and Frequent Genetic Exchange in Leishmania tropica Isolates from Afghanistan, Iran and Syria," has been formally accepted for publication in PLOS Neglected Tropical Diseases.

Best regards,

Shaden Kamhawi

co-Editor-in-Chief

Paul Brindley

co-Editor-in-Chief
